# Laparoscopic Surgery with Concomitant Hernia Repair and Cholecystectomy: An Alternative Approach to Everyday Practice

**DOI:** 10.3390/diseases11010044

**Published:** 2023-03-03

**Authors:** Paul Zarogoulidis, Aris Ioannidis, Marios Anemoulis, Dimitrios Giannakidis, Dimitris Matthaios, Konstantinos Romanidis, Konstantinos Sapalidis, Lavrentios Papalavrentios, Isaak Kesisoglou

**Affiliations:** 13rd Department of Surgery, “AHEPA” University Hospital, Medical School, Aristotle University of Thessaloniki, 54453 Thessaloniki, Greece; 2Surgery Department, Genesis Private Hospital, 54301 Thessaloniki, Greece; 3Surgery Department, General Clinic Euromedica, 54645 Thessaloniki, Greece; 41st Department of Surgery, Attica General Hospital “Sismanogleio-Amalia Fleming”, 57889 Athens, Greece; 5Oncology Department, General Hospital of Rhodes, 86775 Rhodes, Greece; 6University Surgery Department, University General Hospital of Alexandroupolis, 68100 Alexandroupolis, Greece; 7Gastroenterologist, Private Cabinet, 55236 Thessaloniki, Greece

**Keywords:** cholecystectomy, laparoscopy, Antisel, hernia, concomitant, surgery

## Abstract

Introduction: Concomitant surgeries have been performed previously in several centers with experience in laparoscopic surgeries. These surgeries are performed in one patient under one operation with anesthesia. Methods: We performed a retrospective unicenter study from October 2021 to December 2021 analyzing patients who underwent laparoscopic hiatal hernia repair with cholecystectomy. We extracted data from 20 patients who underwent hiatal hernia repair together with cholecystectomy. Grouping of data by hiatal hernia type showed 6 type IV hernias (complex hernia), 13 type III hernias (mixed type) and 1 type I hernia (sliding hernia). Out of the 20 cases analyzed, 19 were patients suffering from chronic cholecystitis and 1 patient presented with acute cholecystitis. The average operating time was 179 min. Minimum blood loss was achieved. Cruroraphy was performed in all cases, mesh reinforcement was added in five cases, and fundoplication was performed in all cases, with 3 Toupet, 2 Dor and 15 floppy Nissen fundoplication procedures performed. Fundopexy was routinely performed in cases of Toupet fundoplication. A total of 1 bipolar and 19 retrograde cholecystectomies were performed. Results: All patients had favorable postoperative hospitalization. Patient follow-up took place at 1 month, 3 months and 6 months, with no sign of recurrence of hiatal hernia (anatomical or symptomatic) and no symptoms of postcholecystectomy syndrome. In two patients, we had to perform colostomy. Conclusion: Concomitant laparoscopic hiatal hernia repair and cholecystectomy is safe and feasible.

## 1. Introduction

Laparoscopic surgery is the gold-standard surgical approach. It can be used for both hiatal hernia repair and cholecystectomy. The main advantage of this surgery is its fast recovery time operatively and postoperatively [1]. There are four hiatal hernia types. Type I hiatal hernias impact quality of life and are refractive to medication. Type II, III and IV symptomatic hiatal hernias are indications for surgery [2]. Some cases require concomitant surgeries. This term refers to performing two or more surgical operations on one patient in one operation. However, with laparoscopic surgeries, there are nowadays few concomitant surgeries. Laparoscopic surgeries have been performed for cholecystectomy [3] and have been the gold standard for hiatal hernia repair during the past 30 years. [2] Laparoscopy is not the standard of care for all surgical interventions, but since it is considered minimally invasive, both hiatus hernia repair and cholecystectomy can be performed as concomitant surgery. Concomitant surgeries under one operation with anesthesia are very appealing, as they have the advantage of minimal invasiveness. Short postoperative stays and early recovery decrease analgesic use. However, there is a prolongation of the operative time, which slightly increases the incidence of intraoperative complications and postoperative morbidities.

## 2. Patients and Methods

We performed a retrospective unicenter study from October 2021 to December 2021, analyzing patients who underwent laparoscopic hiatal hernia repair plus cholecystectomy. We extracted data from patients treated at the 3rd Surgery Clinic of University General Hospital of Thessaloniki and identified 20 patients who underwent hiatal hernia repair and cholecystectomy. Grouping of data by hiatal hernia type showed 6 type IV hernias (complex hernia), 13 type III hernias (mixed type) and 1 type I hernia (sliding hernia). Out of the 20 cases analyzed, 19 were patients suffering from chronic cholecystitis and 1 patient presented with acute cholecystitis. Average operating time was 179 min. Low-molecular-weight heparin (LMWH) and anti-foaming agents were administered to all patients for deep venous thrombosis (DVT) and pulmonary embolism (PE) prophylaxis and bloating reduction. All patients were placed in the Trendelenburg position with both legs separated and both arms tucked along the upper body. The surgeon was standing between the legs, the assistant was on the left side, and the scrub nurse was standing on the right side of the patient. A Hasson approach at the umbilicus for initial placement of a 12 mm trocar was favored, through which pneumoperitoneum at 10–12 mmHg using CO_2_ gas was established. The working trocar was placed in a standard position to triangulate the target organ subcostally in the left anterior axillary line. The retracting trocars were placed subcostally in the right anterior axillary line and in the subxiphoid area.

Minimum blood loss was achieved. Cruroraphy was performed in all cases, mesh reinforcement was added in 5 cases and fundoplication was performed in all cases, with 3 Toupet, 2 Dor and 15 floppy Nissen fundoplication operations. Fundopexy was routinely performed in cases of Toupet fundoplication. We decided to use a synthetic mesh for reinforcement to decrease recurrence in 3 patients with a diaphragmatic defect that resulted from a relaxing incision made for a difficult hiatus and in 2 patients with recurrent hiatal hernia. Figure 1 and Table 1.

## 3. Results

The mean operating time was 155 min. We did not encounter any serious blood loss. Cruroraphy was performed in all cases; in addition, mesh reinforcement was performed in five cases, and fundoplication was performed in all cases, with 2 Toupet, 3 Dor and 14 floppy Nissen operations. Wrap height varied between 1 and 4 cms. Moreover, fundopexy was performed in cases of Toupet fundoplication. Regarding the cholecystectomies, 18 were retrograde and 2 bipolar. Drainage of the gallbladder was performed in 14 cases for 4 days. Nasogastric aspiration tubes were not used. Only two patients were admitted to the ICU. All patients experienced mobilization by postoperative day one, passage of flatus by postoperative day two and had a stool emission by discharge. Postoperative discharge took an average of 5.5 days to occur. Opioids were only used during anesthesia, plus broad-spectrum intravenous antibiotics intraoperatively. Only one patient with acute cholecystitis received broad-spectrum intravenous antibiotics for 6 days and was discharged on the 9th day after surgery. Pain management was achieved using non-opioid analgesia when requested by the patient. Pain was measured preoperatively using the pain visual analogue scale (VAS). Patient follow-up was performed in our outpatient clinic and ranged from 1 month to 6 months, with no patients showing signs of recurrence of hiatal hernia. Five patients had symptoms of postcholecystectomy syndrome, such as fullness or dullness in the upper-right quadrant, especially after movement.

## 4. Discussion

Due to the increased prevalence of laparoscopic surgery and the ability to perform more than one surgery under one operation with anesthesia, more and more centers are performing more than one surgery in one patient, when indicated. The two main surgeries that are performed as one surgery are hiatal hernia repair and cholecystectomy through a laparoscopic approach. Again, this is feasible and safe, and performing both surgeries under one operation with anesthesia precludes the need for a second hospitalization. The quality of life is improved and the patient has no need to take multiple work breaks [4]. One of the advantages of this surgery is that there is no need for a second trocar insertion when hiatal hernia repair is performed, as the trocar positioning resembles the French technique used for cholecystectomy. This concomitant surgery should only be performed by an experienced surgeon who regularly performs hiatal hernia repair. Obesity is not a negative predictive factor for a concomitant surgery. The main reason for performing a concomitant surgery is that both surgeries are necessary. We recommend the trocar insertion be planned beforehand, keeping in mind ergonomics and triangulation in laparoscopic knot tying and suturing. In morbidly obese patients, the umbilicus is not a reliable landmark and the landmark should be positioned 15 cm from the xiphoid on the xipho-umbilical line. Imaging plays a crucial role and should always be performed before such surgeries. Upper digestive endoscopy should be performed in order to diagnose hiatal hernia as well as esophagitis. Cameron ulcers and other benign or malign coexisting pathologies can lead to a decision for surgery. Parahiatal hernia can be confirmed via barium meal [5]. Barium meal is the gold standard for diagnosing hiatus hernia, and it has the benefit of excellent visualization of the hernia sac. Barium meal was not performed in all patients since we were able to classify some hernias with only upper endoscopy, since they were large. Barium meal provides useful information regarding the herniated stomach, but is lacking in cases of complex hernias in which other abdominal organs are herniated. Computed tomography (CT) should be performed prior to every surgery as it provides the most information, especially if enhanced with intravenous and oral contrast. It can diagnose gallbladder stones, choledocholithiasis and all types of hiatus hernias. Moreover, it provides the particular anatomy of every patient. Currently, tridimensional reconstruction software and preoperative planning software based on CT are useful tools for the proper surgical planning of trocar positioning. Based on this information, we achieved better ergonomics for the surgeon, decreased blood loss and reduced operating time. Performing both surgeries is easier and is both safe and feasible in the hands of an experienced laparoscopic team. Our mean operating time was 170 min compared to the average operating time of standalone hiatal hernia repairs, which was 130 min. The operating time was similar when compared with standalone surgery and minimum blood loss was achieved in both cases. Oral intake along with bowel movement and active mobility resumption was similar in standalone hiatal hernia surgery. However, although evidence exists regarding the safety of placing mesh in clean-contaminated wounds [6,7,8,9,10,11], it is our opinion that the current evidence is not sufficient, and we advise the use of mesh only in carefully selected cases if cholecystectomy is performed. Meshes might induce serious complications, and we believe that it should only be used as a last resort in selected recurrent hiatal hernia patients or in patients with a diaphragmatic defect that resulted from a relaxing incision for a difficult hiatus, as in this study [12]. We recommend firstly performing the hiatal hernia and then cholecystectomy. It has been observed that for surgeons who operate between the patients’ legs, the transition is easy; however, for surgeons who perform hiatal hernia repair from the right side of the patient, the transition is more difficult. In this case, surgeons have to change the position of the laparoscope. Achilles heel and bile duct injuries commonly occur during laparoscopic surgery versus open surgery, with reported rates being higher in laparoscopic operations [11,13,14,15,16,17,18]. Moreover, we should take into account the fatigue that follows for the surgeon. Laparoscopic cholecystectomy is a relatively common and simple procedure; however, since the surgeon is performing it after a hiatal hernia repair, fatigue might be observed. The surgeon should be in a familiar position for cholecystectomy, and should bear in mind the risk of this injury, especially due to fatigue. Since two procedures are performed, two different intraoperative incidents may occur; however, we did not observe a higher rate of morbidity. This was due to our experience; we believe that an experienced center should have more than 30 double surgeries per year, making use of our excellent ergonomic tools. Regarding the two necessary colostomies, these were performed due to errors made by our residents during their surgical training. Moreover, in the case of suspicion of an adverse effect, a CT of the upper and/or lower abdomen should be performed. In conclusion, concomitant laparoscopic hiatal hernia repair and cholecystectomy is a safe and feasible option for patients with indication of surgery for both pathologies.

## Figures and Tables

**Figure 1 diseases-11-00044-f001:**
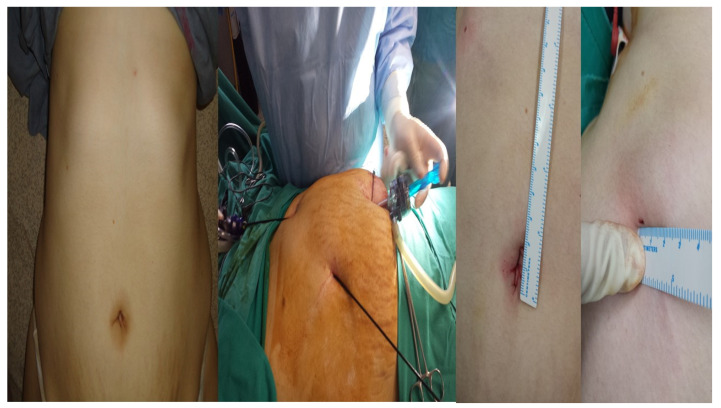
From left to right. The patient before surgery, trocars and equipment placement during surgery, traumatic postsurgery holes from the equipment. A total of 1 bipolar and 19 retrograde cholecystectomies were performed.

**Table 1 diseases-11-00044-t001:** Tools used.

-evoPOUCH Specimen Retrieval Bag
-evoPORT Family of Trocars
-evoREACH Endoscopic Instruments
-evoFLEX Linear Cutter Stapler
-evoPNEU INSUFLATION NEEDLES
-evoLUTION LINEAR CUTTER STAPLER V series
-evoLUTION LINEAR STAPLER V series
-evoMED Family of Staplers
-evoMED evoclip ligating system
-evoMED evolapse V series
-evolution circular stapler V series

See Appendix A.

## Data Availability

All data are available if requested by the corresponding author.

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
