# Peer review of "Laparoscopic Surgery with Concomitant Hernia Repair and Cholecystectomy: An Alternative Approach to Everyday Practice"

_diseases, 2023, doi:10.3390/diseases11010044_

Round 1

Reviewer 1 Report

Seemingly interesting paper regarding the possible advantages of combined surgical approach, it has some relevant flaws:

1. No results session is reported

2. It is unclear in methods when Authors decide to use mesh: is it size of defect? recurrence, as cited in Discussion

3. Something better regarding  surgical approach (trocar and staff position should be reported), as the focus of the paper regards new "tricks"

From this point of view, may be a couple of figures could make it clearer (e.g. trocars and staff position) 

Author Response

Reviewer 1

  1. No results session is reported

Answer

Thank you for your comment

We have added the following section

Results

Mean operating time was 155 minutes. In did not encounter any serious blood loss. Cruroraphy was performed in all cases, in addition mesh reinforcement was performed in five cases, and fundoplication was added in all cases. 2 Toupet, 3 Dorr and 14 Floppy-Nissen. Wrap height varied between 1–4 cms. Moreover; fundopexy was added in cases with Toupet fundoplication. Cholecystectomy was performed 18 retrograde, 2 bipolar. Drainage for gallbladder was placed added in 14 cases for 4 days. Nasogastric aspiration tube was not used. Only two patients where admitted in the ICU. All patients started mobilization by postoperative day one, passage of flatus by postoperative day two and had a stool emission by discharge. Postoperative discharge was an average of 5.5 days. Opioids were only used during anesthesia, plus broad spectrum intravenous antibiotic intraoperatively. Only  one patient with acute cholecystitis received broad spectrum intravenous antibiotic 6 days and was discharged on the 9th day after surgery. Pain management was achieved by non-opioid analgesia on demand by the patient. Pain was measured preoperatively using the Pain visual analogue scale (VAS) preoperative. Patient follow-up was performed in our outpatient cabinet ranging from 1 month to 6 months, with no sign of recurrence for hiatal hernia. Five patients had postcholecystectomy syndrome, like fullness or dullness in the upper right quadrant, especially after some movements.

We have added this section and highlighted in yellow

  1. It is unclear in methods when Authors decide to use mesh: is it size of defect? recurrence, as cited in Discussion

Answer

Thank you for your comment

In the patients and methods section we have added the following information which is now highlighted in yellow.

It was decided to use a synthetic mesh for reinforcement to decrease recurrence in 3 patients with a diaphragmatic defect that resulted from a relaxing incision for a difficult hiatus and in 2 patients recurrent hiatal hernia.

  1. Something better regarding  surgical approach (trocar and staff position should be reported), as the focus of the paper regards new "tricks"

Answer

Thank you for your comment

We have added this information in the Methods section and highlighted the corrections in yellow.

Low molecular weight heparin (LMWH) and anti-foaming agents were administered to all patients for deep venous thrombosis (DVT) and pulmonary embolism (PE) prophylaxis and bloating reduction. All patients were placed in Trendeleburg position with both legs separated and both arms tucked along the upper body. The surgeon was standing between the legs, the assistant was on the left side and the scrub nurse was standing on the right side of the patient. A Hasson approach at the umbilicus for initial placement of a 12 mm trocar was favored through which a pneumoperitoneum at 10–12 mmHg by CO2 gas was established. Working trocar was placed in a standard design used to triangulate the target organ, subcostally in the left anterior axillary line. The retracting trocars were placed subcostally in the right anterior axillary line and in the subxiphoid area.

Also, the title was changed to Laparoscopic Surgery Concomitant Hernia Repair and Cholocystectomy an alternative approach to everyday practice

From this point of view, may be a couple of figures could make it clearer (e.g. trocars and staff position) 

Answer

We do not have such figures to add we are sorry this

Reviewer 2 Report

Dear authors, thank you for sending your article to our  journal. In this manuscript you analize your experience performing concomitant surgeries, laparoscopic hiatal hernia repair and colecistectomy.

The manuscript could be improved. Methods are not clear, items to define surgery safety are not  described. Results should be described in a most extensively.

Finally, figures in supplementary file could be improved.

Author Response

Reviewer 2

The manuscript could be improved. Methods are not clear, items to define surgery safety are not  described. Results should be described in a most extensively.

Finally, figures in supplementary file could be improved.

Answer

Thank you for your comment

The following have been added and highlighted in yellow

Patients and methods

Low molecular weight heparin (LMWH) and anti-foaming agents were administered to all patients for deep venous thrombosis (DVT) and pulmonary embolism (PE) prophylaxis and bloating reduction. All patients were placed in Trendeleburg position with both legs separated and both arms tucked along the upper body. The surgeon was standing between the legs, the assistant was on the left side and the scrub nurse was standing on the right side of the patient. A Hasson approach at the umbilicus for initial placement of a 12 mm trocar was favored through which a pneumoperitoneum at 10–12 mmHg by CO2 gas was established. Working trocar was placed in a standard design used to triangulate the target organ, subcostally in the left anterior axillary line. The retracting trocars were placed subcostally in the right anterior axillary line and in the subxiphoid area.

It was decided to use a synthetic mesh for reinforcement to decrease recurrence in 3 patients with a diaphragmatic defect that resulted from a relaxing incision for a difficult hiatus and in 2 patients recurrent hiatal hernia.

Results

Results

Mean operating time was 155 minutes. In did not encounter any serious blood loss. Cruroraphy was performed in all cases, in addition mesh reinforcement was performed in five cases, and fundoplication was added in all cases. 2 Toupet, 3 Dorr and 14 Floppy-Nissen. Wrap height varied between 1–4 cms. Moreover; fundopexy was added in cases with Toupet fundoplication. Cholecystectomy was performed 18 retrograde, 2 bipolar. Drainage for gallbladder was placed added in 14 cases for 4 days. Nasogastric aspiration tube was not used. Only two patients where admitted in the ICU. All patients started mobilization by postoperative day one, passage of flatus by postoperative day two and had a stool emission by discharge. Postoperative discharge was an average of 5.5 days. Opioids were only used during anesthesia, plus broad spectrum intravenous antibiotic intraoperatively. Only  one patient with acute cholecystitis received broad spectrum intravenous antibiotic 6 days and was discharged on the 9th day after surgery. Pain management was achieved by non-opioid analgesia on demand by the patient. Pain was measured preoperatively using the Pain visual analogue scale (VAS) preoperative. Patient follow-up was performed in our outpatient cabinet ranging from 1 month to 6 months, with no sign of recurrence for hiatal hernia. Five patients had postcholecystectomy syndrome, like fullness or dullness in the upper right quadrant, especially after some movements.

We changed the title to

Laparoscopic Surgery Concomitant Hernia Repair and Cholocystectomy an alternative approach to everyday practice

Unfortunately

We do not have any additional figures

Round 2

Reviewer 1 Report

Authors addressed most reviewer's points (Fig of staff and trocar positioning is the only missing one)

Author Response

We have added firgure number 1 to our manuscript
